# Cytotoxic Efficacy and Resistance Mechanism of a TRAIL and VEGFA-Peptide Fusion Protein in Colorectal Cancer Models

**DOI:** 10.3390/ijms22063160

**Published:** 2021-03-19

**Authors:** Michal Kopczynski, Malgorzata Statkiewicz, Magdalena Cybulska, Urszula Kuklinska, Katarzyna Unrug-Bielawska, Zuzanna Sandowska-Markiewicz, Aleksandra Grochowska, Marta Gajewska, Maria Kulecka, Jerzy Ostrowski, Michal Mikula

**Affiliations:** 1Department of Genetics, Maria Sklodowska-Curie National Research Institute of Oncology, 02-781 Warsaw, Poland; michal.kopczynski@pib-nio.pl (M.K.); malgorzata.statkiewicz@pib-nio.pl (M.S.); magdalena.cybulska@pib-nio.pl (M.C.); ukuklinska@gmail.com (U.K.); katarzyna.unrug-bielawska@pib-nio.pl (K.U.-B.); zuzanna.sandowska-markiewicz@pib-nio.pl (Z.S.-M.); grochowska.am@gmail.com (A.G.); marta.gajewska@pib-nio.pl (M.G.); mkulecka@cmkp.edu.pl (M.K.); jostrow@warman.com.pl (J.O.); 2Centre for Postgraduate Medical Education, Hepatology and Clinical Oncology, Department of Gastroenterology, 01-813 Warsaw, Poland

**Keywords:** TRAIL, colorectal cancer, PDX, apoptosis

## Abstract

TNF-related apoptosis-inducing ligand (TRAIL) is a type II transmembrane protein capable of selectively inducing apoptosis in cancer cells by binding to its cognate receptors. Here, we examined the anticancer efficacy of a recently developed chimeric AD-O51.4 protein, a TRAIL fused to the VEGFA-originating peptide. We tested AD-O51.4 protein activity against human colorectal cancer (CRC) models and investigated the resistance mechanism in the non-responsive CRC models. The quantitative comparison of apoptotic activity between AD-O51.4 and the native TRAIL in nine human colorectal cancer cell lines revealed dose-dependent toxicity in seven of them; the immunofluorescence-captured receptor abundance correlated with the extent of apoptosis. AD-O51.4 reduced the growth of CRC patient-derived xenografts (PDXs) with good efficacy. Cell lines that acquired AD-O51.4 resistance showed a significant decrease in surface TRAIL receptor expression and apoptosis-related proteins, including Caspase-8, HSP60, and p53. These results demonstrate the effectiveness of AD-O51.4 protein in CRC preclinical models and identify the potential mechanism underlying acquired resistance. Progression of AD-O51.4 to clinical trials is expected.

## 1. Introduction

Apoptosis is a type of programmed cell death that plays a crucial role in the preservation of tissue homeostasis by maintaining an adequate cell population [1]. Two apoptosis-inducing pathways have been identified in mammalian cells [2]. The extrinsic pathway is activated by tumor necrosis factor (TNF) receptor superfamily members upon binding to the corresponding ligand [3]. This leads to the recruitment of death-inducing signaling complex (DISC) proteins to propagate the apoptotic signal [4], resulting in Caspase-8 activation [5] and the degradation of key structural proteins [6]. The intrinsic pathway, referred to as the mitochondrial pathway, is initiated by non-receptor-mediated stimuli such as radiation, DNA damage, free radicals, toxins, or hypoxia [7]. These stimuli lead to the mitochondrial permeabilization and the release of apoptosis-inducing factors such as cytochrome C or Smac/DIABLO [8,9]. Cytochrome C binds to Apaf-1, forming a complex that activates Caspase-3 via Caspase-9 [10]. Smac/DIABLO neutralizes inhibitor of apoptosis proteins (IAPs), leading to Caspase-9 activation [11].

TNF-related apoptosis-inducing ligand (TRAIL) is a type II transmembrane protein [12] that is cleaved by proteases to release the soluble form [13]. Unlike other ligands of the TNF family, TRAIL selectively induces apoptosis in cancer cells [14]. To induce apoptosis, TRAIL (as a homotrimer) binds to its cognate death receptors (DRs) [15]. Four membrane-bound TRAIL receptors have been described, including TRAIL-R1/TNFRSF10A [16], TRAIL-R2/TNFRSF10B [17], TRAIL-R3/TNFRSF10C/DcR1 [18], and TRAIL-R4/TNFRSF10D/DcR2 [19]. Throughout this paper, the shortest naming system (R1, R2, DcR1, DcR2) will be used. TRAIL also binds with low affinity to the soluble osteoprotegerin receptor [20]. R1 and R2 have a conserved and fully functional death domain (DD), and upon ligand binding, they induce DISC formation and activate the Caspase cascade, leading to cell death [21]. The other two membrane-bound receptors are called decoy receptors; this is because they cannot propagate the death signal due to truncated (R4) or deficient (R3) DD regions [22]. Although TRAIL-mediated apoptosis is mainly mediated by the extrinsic pathway, the apoptotic signal can be strengthened by mitochondrial pathway activation [23]. In most cells, the amount of Caspase-8 is sufficient for Caspase-3 activation. However, in cells with low Caspase-8 levels, the apoptotic signal can be amplified [24] by the proteolytic activation of the Bid protein [25], resulting in mitochondrial membrane permeabilization. One of the strongest principles of TRAIL-mediated apoptosis is independence from the p53 protein, which is mutated in at least 50% of cancers [26]. TRAIL-induced apoptosis is tightly regulated, being a positive factor in comparison to other members of the TNF superfamily. However, such tight regulation could be associated with vulnerability to deregulation at various checkpoints. Resistance to TRAIL is defined according to the origin: it can be external (associated with changes in DRs) or internal (characterized by alterations in DISC and Caspase cascade activity).

The exploitation of the TRAIL ligand–receptor system has been pursued as a promising therapeutic approach for the preferential cancer cells targeting. However, to date, despite proven safety and tolerability at clinical trials, neither recombinant forms of TRAIL nor antibodies against TRAIL-R1/R2 showed a favorable clinical response in cancer patients, mainly due to the resistance mechanism [27]. AD-O51.4 is a TRAIL-based ligand expressed as a fusion protein with vascular endothelial growth factor VEGF-derived peptides, strongly inhibiting tumor growth and capable of bypassing TRAIL resistance, as well as of inhibiting angiogenesis [28]. Here, using in vitro and in vivo colorectal cancer (CRC) models, we demonstrated the promising therapeutic effects of AD-O51.4 protein, and proposed a potential mechanism underlying the resistance of CRC cells acquired after prolonged exposure to AD-O51.4.

## 2. Results

### 2.1. The AD-O51.4 Inhibits Growth of CRC Cell Lines (In Vitro and In Vivo) and Patient-Derived Xenografts (PDXs)

The cytotoxic effect of AD-O51.4 on CRC cell lines was assessed using the MTT assay. Cells were treated in parallel with either AD-O51.4 or the native TRAIL protein to compare toxicity profiles. AD-O51.4 showed similar or better cytotoxic activity than TRAIL (Figure 1), and the effect was dose-dependent. The tested compound showed high (IC_50_ < 0.1 nM) and moderate (IC_50_ > 0.1 nM) toxicity against Colo-205, DLD-1, CL-11, SW-480 cells and HCT-116, CL-40, HT29 cells, respectively, while the RKO and CACO-2 cell lines showed no response to AD-O51.4 and TRAIL. To validate the species specificity of AD-O51.4, the same experiments were performed using three mouse cell lines. Neither TRAIL nor AD-O51.4 showed toxic activity against the B-16 cell line (melanoma), whereas AD-O51.4 exhibited low toxicity toward mIMCD3 (kidney inner medullary collecting duct cells) and CT-26 (colon carcinoma) cells indicating that human ligand proteins to not interact or bind loosely with mouse TRAIL receptor. In mouse, two decoy receptors and one death-mediating TRAIL receptor with highest homology to human TRAIL-R2 have been reported [29]. Although the toxic activity was detected only at the highest concentration (10 nM), AD-O51.4 was considered specific and safe for further in vivo studies using immunocompromised animals.

To further examine the potential of AD-O51.4 as a therapeutic agent, in vivo studies were performed using xenograft models in NU/J mice. Four CRC cell lines (CACO-2, HCT-116, HT-29, and Colo-205), which showed different responses to AD-O51.4 treatment in the in vitro experiments, were employed. AD-O51.4 delayed, although not statistically significantly, tumor growth in the HT-29 model, arrested tumor growth in the CACO-2 and HCT116 models, and decayed tumors in Colo-205 xenografts. These results showed that AD-O51.4 inhibited CRC xenograft tumor growth with high efficacy, including the CACO-2 xenograft which was resistant in vitro (Figure 2). Similar experiments were performed using five patient-derived xenograft (PDX) models, which were recently genetically and transcriptionally characterized by our group [30], and represented three out of four consensus molecular subtypes (CMSs) of the CRC [31]. Four PDX models showed a reduction in tumor volume after the AD-O51.4 treatment. Growth arrest was observed in one case (X29) (Figure 2). No significant body weight loss/increase was detected in response to treatment, suggesting that AD-O51.4 had no unwanted side effects. To relate to the discrepancy of CACO-2 in vitro (resistant) and in vivo (responsive) findings, using this cell line we performed an additional in vivo experiment with TRAIL treatment as an additional variable. The AD-O51.4 turned out to be significantly more efficient in blocking CACO-2 xenograft growth, probably owing the impact of VEGFA-derived effector peptides (Figure 3).

### 2.2. The Sensitivity of CRC Cell Lines to AD-O51.4 Correlates with Cell Surface Expression of TRAIL Receptors

To determine the role of TRAIL receptors in AD-O51.4-induced apoptosis, surface receptors were quantified in cell lines by immunofluorescence staining (Figure 3A,B). We found this approach more accurate than Western blotting as the actionable ligand–receptor interaction occurs on to the cell surface. The raw comparison of the cell surface signals derived from specific receptors can lead to misinterpretation due to differences in the size and shape of cell lines. Therefore, we measured the signal strength derived from actin stained with phalloidin to normalize the signal. This enabled the measurement of the abundance of each cell surface TRAIL receptor, and normalization to actin provided data about receptor density rather than raw amounts. The DLD-1 cell line, which was highly responsive to AD-O51.4, expressed high levels of both proapoptotic R1 and R2 as well as decoy DcR1 and DcR2 receptors (Figure 4A), whereas the resistant CACO-2 cell line expressed low levels of all receptors, suggesting a relationship between expression levels and resistance (Figure 4B). The exception was the RKO cell line that exhibits high levels of R1, R2, and DcR1 receptors, at the same time being TRAIL/AD-O51.4 resistant. The correlation of IC_50_ data and relative values of R1 and R2 receptors abundances for seven sensitive cell lines showed significant inverse correlation of R2 receptor abundance alone (Pearson coefficient = −0.76; *p* value = 0.047), R1 and R2 combined (Pearson coefficient = −0.87; *p* value = 0.01), but not for R1 receptor alone (Pearson coefficient = −0.69; *p* value = 0.087). The correlation of individual DcR1 and DcR2 receptors abundances with IC_50_ data was not significant (Pearson coefficient = −0.58; *p* value = 0.1702 and Pearson coefficient = −0.54; *p* value = 0.2094, respectively), however, it was significant for decoy receptor abundances combined (Pearson coefficient = −0.82; *p* value = 0.0238) and all TRAIL receptors abundances total (Pearson coefficient = −0.85; *p* value = 0.015) (Figure 4C). Overall, these results showed that the extent of cell death correlates with cumulative TRAIL receptors abundance on the panel of tested CRC cell lines.

### 2.3. AD-O51.4-Induced Resistance Is Mediated by Both a Reduction in the Expression of the Cell Surface Proapoptotic TRAIL Receptor and Changes in Expression of Apoptosis-Related Proteins

To further examine the mechanisms underlying acquired resistance to AD-O51.4, two cell lines (SW-480 and DLD-1) were cultured in the presence of increasing doses of AD-O51.4 to generate resistant cells. Cells were exposed to 0.05 nM AD-O51.4 (corresponding to the IC_50_) and treatment was continued until resistance at least ten times higher than that of the primary cell line was achieved (Figure 5A). Comparable MTT experiments were performed using the native TRAIL protein to determine whether resistance was AD-O51.4-specific. The cell lines demonstrated similar resistance to both TRAIL and AD-O51.4. Moreover, acquired resistance is preserved after xenograft implantation (Figure 6) suggesting that the resistance is stable and long-lasting. Moreover, we observed the time reduction (8 vs. 12 days in the case of native DLD-1) needed to reach a starting tumor volume for AD-O51.4 administration. Such observation could suggest a link between AD-O51.4 acquired resistance and an increase in tumor aggressiveness, but this result needs further confirmation.

The quantitative analysis of immunofluorescence staining data revealed a significant (*p* value < 0.05) downregulation of proapoptotic (R1 and R2) receptor abundance in both AD-O51.4 resistant cell lines compared with the parental AD-O51.4-sensitive cell lines. Furthermore, the DcR1 decoy receptor was downregulated in the SW-480 AD-O51.4-resistant cell line (Figure 5B,C). This result indicated a link between AD-O51.4/TRAIL resistance and the depletion of TRAIL receptors. Because this approach is limited to the cell surface, we then measured the expression of protein related to apoptosis; for this we used a Human Apoptosis Antibody Array accommodating 43 apoptotic markers. The quantitative analysis of densitometry measurements identified 19 and 17 significantly altered proteins in AD-O51.4-resistant cells cultured in the presence and absence of AD-O51.4, respectively (Figure 7). A similar experiment using native versions of mentioned cell lines could not be performed due to the rapid cell death at the given AD-O51.4 concentrations. Compared with the immunofluorescence data, there was no decrease in the expression of TRAIL R1 and R2 receptors; however, there was a significant increase in DcR1 and DcR2 receptors, which contrasts with the immunofluorescence data. Importantly, the AD-O51.4-resistant cells demonstrate a significant reduction in the abundance of Caspase-8, thereby creating a roadblock to an appropriate propagation of apoptotic signals. Furthermore, we observed a significant reduction in HSP60 and p53, which constitute not only a mitochondrial apoptosis inhibitor complex [32], but also Caspase-3 maturation [33]. Taken together, these data show that a possible relocation of TRAIL receptors from the cell surface, along with the concomitant depletion of Caspase-8 and a low p53 abundance, could be ultimately responsible for acquired AD-O51.4 resistance in CRC cell lines.

## 3. Discussion

Despite progress in cancer biology and the development of novel therapeutic strategies, cancer remains one of the principal causes of death worldwide. Surgical resection and chemo/radiotherapy are the most common methods for the treatment of cancer. However, these therapies are limited by a lack of specificity, which results in damage to healthy tissues. The development of novel targeted treatments for cancer will hopefully lead to improved results in the future [34].

TNF, which was first identified in 1975, is a death ligand used for anticancer therapy [35]. Despite initial promising results, TNF-α treatment is associated with severe toxicities [36]. Another cancer-specific apoptosis-inducing receptor/ligand pair is Fas/FasL [37], however, it met the same fate, as it caused lethal hepatotoxicity [38]. A promising step was the discovery of another TNF superfamily member, TRAIL [39], which differs from other family members in its ability to induce apoptosis in tumors without toxicity toward normal cells [40]. This discovery stimulated the development of clinically usable TRAIL receptor agonists and promoted studies on TRAIL-mediated apoptosis. However, resistance to TRAIL is a major limitation with respect to clinical use [41]. However, research and understanding about molecular alterations that lead to acquired resistance during prolonged TRAIL exposure are necessary for the success of TRIAL-based human cancer treatment. To meet this challenge, we must not only examine the apoptotic activity of novel TRAIL-like proteins, but also identify possible alterations through which cancer cells develop resistance to TRAIL-like proteins.

Here, we evaluated the effects of AD-O51.4, a TRAIL-like protein, in vitro and in vivo. In contrast to the paper of Rozga et al. [28], which has more screening character, here we focused only on AD-O51.4’s anticancer activity towards human colorectal cancer. Our study not only incorporated various CRC models like established cell lines, cell line-derived xenografts or patient-derived xenografts, but also revealed a possible mechanism of acquired resistance.

AD-O51.4 was more toxic to human CRC cell lines than the native TRAIL protein. This effect was observed in seven out of nine cell lines; RKO and CACO2 were resistant to both TRAIL and AD-O51.4. AD-O51.4 maintained its predecessor’s species specificity; it showed no toxicity toward three mouse cell lines. These data provided the basis for further examination of AD-O51.4 as a potential biological therapeutic in a xenograft model.

AD-O51.4 delayed the growth of tumors derived from four out of the five cell lines tested, even the CACO2 cell line that was resistant in vitro. Similar experiments using PDXs showed tumor decay in four cases and growth arrest in one case, with no negative side effects. Comparable results have been reported recently by Rozga et al. using pancreatic, gastric, lung, and colon carcinomas [28].

The significant correlation of the IC_50_ data and receptor abundances for seven sensitive CRC cell lines suggests that the extent of TRAIL-induced cell death may be related to the abundance of cell surface receptors. Indeed, alterations in the expression of R1 and R2 receptors directly impact the strength of apoptotic signal propagation [42]. Moreover, levels of DcR1 and DcR2 are crucial [43,44]. This is due to the fact that decoy receptors not only disrupt signal propagation by forming non-functional receptor heterocomplexes, but also compete with R1 and R2 for the TRAIL ligand [45]. From the above and our data (Figure 1 vs. Figure 4C supported by Figure 5C), we propose that the crucial matter is not the level of individual cell surface receptors but rather their abundance ratios. For instance, the CACO-2 receptor pattern is characterized by significantly lower R1 and R2 levels than other tested cell lines. Thus, we conclude that it lacks sufficient positive relays for signal propagation. By contrast, RKO is characterized by high levels of R1 and R2, and high levels of DcR1. Given the fact that CL-40 or Colo-205 showed a comparable receptor abundance ratio, but remained sensitive to AD-O51.4, we suggest that RKO resistance has intracellular origins. From the above, we propose that surface-related TRAIL receptor abundance acts as a first step biological marker for predicting responses to TRAIL-like CRC treatment. CACO-2 and RKO cases show that resistance to TRAIL-like proteins could be multi-origin, which implies a need for a combined anticancer approach.

We addressed the mechanisms underlying resistance originating from the long-term administration of TRAIL-like proteins in CRC cell lines. The immunofluorescence studies confirmed the decreased expression of R1 and R2 cell surface receptors by cell lines with AD-O51.4-acquired resistance without significant changes in the expression of DcR1 and DcR2. The depletion of proapoptotic TRAIL receptors is observed in other types of cancer cell [46,47], suggesting that their depletion is a common resistance mechanism. Further investigation of the total protein levels revealed the unchanged expression of unchanged R1 and R2 along with a significant increase in the expression of DcR1 and DcR2. The disparity between these two readouts could be attributed to the alteration in the distribution of TRAIL receptors via protein secretion and endocytosis pathways [48], leading to the increased localization in the cytosol or nuclear perimeter [49]. Such phenomena (despite proper receptor synthesis) could lead to reduced levels of cell surface receptor, thereby evoking weaker reactions to TRAIL-like ligands [50]. This leads to the conclusion that not only the cell surface expression levels, but also the relocalization of TRAIL receptors to the cell membrane, determine TRAIL sensitivity.

Many tumor types show internally sourced TRAIL resistance, which is connected to alterations in distinct protein families [51,52,53]. For example, the binding of C-FLIP, which contains two death effector domains (similar to Caspase-8) [54], but lacks an active catalytic domain [55], inhibits Caspase-8 activation [56]. Increased expression of c-FLIP correlates with the degree of malignancy in many tumors [57,58], as well as resistance to TRAIL [59,60]. In cells with a predominance of the mitochondrial pathways, the Bcl-2 family of proteins, which includes pro- and antiapoptotic proteins, plays a prominent role. Among proapoptotic proteins, Bak/Bax plays a crucial role in mitochondrial membrane permeabilization [61], and its downregulation correlates with resistance to TRAIL [62,63]. By contrast, the overexpression of the antiapoptotic Bcl-XL or Bcl-2 proteins protects cells from TRAIL-mediated apoptosis and promotes cell survival [64,65] by inhibiting Bax-mediated mitochondrial permeabilization. Moreover, resistance to TRAIL can be regulated by IAPs via the inhibition of Caspase-9 activation [66], and TRAIL-resistant cells express high levels of IAPs [67]. Moreover, Smac/DIABLO proteins, which induce IAPs and restore TRAIL sensitivity, are found at low levels in human melanoma [68]. In our study, we examined the fold changes in the expression of 43 TRAIL-induced apoptosis-related proteins. We propose a mechanism by which CRC acquires resistance to TRAIL-like protein-induced apoptosis after prolonged treatment (Figure 5). We detected the rare activation of Caspase-3 that was independent of Caspase-8. Such a phenomenon was also observed by Feng [69], who associated it with an overproduction of Fas and FasL, which we also observed (Figure 8); this can be explained by AD-O51.4-Fas activation or Fas mutation-induced autoantibody production [70].

Despite extensive research, there are a few potential TRAIL-based therapeutic agents for the treatment of cancer. Although many of these showed promising results in vitro and in vivo, they failed to show efficacy in clinical studies [71] or were associated with increased hepatotoxicity [72]. The most advanced TRAIL-like potential drug reached a Phase II in a clinical study in 2018 [73]; however, the results were not promising [74,75]. Although the development of TRAIL-like agents for targeted cancer treatment was initially encouraging, the end results have so far been unsatisfactory. In the present study, we show that the AD-O51.4 chimeric protein is an effective anti-CRC molecule in preclinical in vitro and in vivo models. Given the recent study showing the efficacy of AD-O51.4 against other cancer types in vivo and the announcement of toxicology studies [28], the progression of AD-O51.4 to clinical trials is expected.

## 4. Materials and Methods

### 4.1. Cell Culture

Cell lines were purchased from American Type Culture Collection and grown in the indicated media supplemented with heat-inactivated fetal bovine serum as follows: DLD-1 (DMEM 10%), CL-11 (DMEM-F12 10%), CL-40 (DMEM-F12 10%), HCT-116 (McCoy 10%), HT-29 (McCoy 10%), SW-480 (RPMI 10%), RKO (MEM 10%), CACO-2 (MEM 20%), Colo-205 (RPMI 10%), B-16 (DMEM 10%), CT-26 (RPMI 10%) and mIMCD-3 (DMEM-F12 10%). Cells were grown in a humidified 5% CO_2_ atmosphere at 37 °C.

### 4.2. Measurement of Cell Viability

Cell viability was measured using the MTT assay. Cells were seeded at two densities of 1 × 10^4^ and 1.5 × 10^4^ cells per well in 96-well plates and maintained in the appropriate culture medium. After 24 h, the cells were cultured in serum-free medium for 24 h, followed by treatment with the indicated concentrations of each compound for 48 h. Cells were then washed with PBS and treated with MTT solution (final concentration, 5 mg/mL) for 3 h at 37 °C. The supernatant was removed, and the formazan crystals were dissolved in 100 µL of isopropanol. Absorbance at 540 nm was measured with a microplate reader VICTOR 3 Model (PerkinElmer, Waltham, MA, USA).

### 4.3. Immunofluorescence Staining

The cells were seeded on 12 mm coverslips in a 24-well plate (7 × 10^4^ cells/well). Cells were starved for 24 h in serum-free medium and then treated with AD-O51.4/TRAIL (10 nM) for the indicated times, rinsed with PBS, and fixed with 3.6% paraformaldehyde in PBS for 15 min at room temperature. Then, the cells were permeabilized and blocked with solution I (0.1% *w*/*v* saponin (S7900, Sigma-Aldrich, Saint Louis, MO, USA), 0.2% gelatin (G7765, Sigma-Aldrich, Saint Louis, MO, USA), and 5 mg/mL BSA in PBS for 10 min at room temperature. Cells were further incubated for 60 min with primary antibodies prepared in solution II (0.01% *w*/*v* saponin and 0.2% gelatin in PBS). The following antibodies were used: TNFRSF10A (AF347, R&D Systems, Minneapolis, MN, USA), TNFRSF10B (AF631, R&D Systems, Minneapolis, MN, USA), TNFRSF10C (AF630, R&D Systems, Minneapolis, MN, USA), and TNFRSF10D (AF633, R&D Systems, Minneapolis, MN, USA). Cells were then incubated with donkey anti-goat secondary antibody conjugated with Alexa Fluor 647 (ab150131, Abcam, Cambridge, UK) (45 min) in solution II. Actin was visualized by staining with phalloidin conjugated with Alexa Fluor 350 (A22281, Thermo Fisher Scientific, Waltham, MA, USA). Z-stack images were acquired with a Zeiss LSM800 microscope (Zeiss, Oberkochen, Germany) equipped with a 63×/1.40 oil immersion objective at 1024 × 1024 pixel resolution and analyzed using ZEN 2009 software (Zeiss, Oberkochen, Germany). Maximum intensity projections of Z-stacks were exported as TIFF files and arranged using Adobe Photoshop CS5.1 software (Adobe, San Jose, CA, USA) with only linear adjustments.

### 4.4. Detection and Quantification of Apoptosis-Related Proteins

Total protein extracts were isolated from native SW-480 and SW-480 cells with acquired AD-O51.4 resistance cultured in the presence or absence of AD-O51.4. Aliquots containing 500 µg of protein were analyzed using the Human Apoptosis Antibody Array following the manufacturer’s recommendations (ab134001, Abcam, UK). Blots were visualized using a chemiluminescence imaging system. Densitometric analysis was performed using Image Studia Lite v5.2 (Li-cor Biosciences, Lincoln, NE, USA). For array-to-array comparisons, densitometry data were normalized to positive control signals of randomly selected blots.

### 4.5. In Vivo Studies

#### 4.5.1. Mice Breeding

NU/J (nude) athymic mice, purchased from The Jackson Laboratory (Bar Harbor, ME, USA), were maintained in an SPF facility under proper environmental conditions (temperature, humidity, and 12 h light cycle) with free access to water and food. The core of the breeding colony was a group of sister and brother (female heterozygote × male homozygote) mated animals kept in our bank of inbred strains.

#### 4.5.2. Experiments with Xenografts

Xenograft implantations were performed on 6–18 week-old mice in a separate operating room using aseptic procedures. Six different colorectal cancer patient-derived xenografts established previously [30] at Maria Sklodowska-Curie National Research Institute of Oncology were implanted subcutaneously into the flanks of six animals. To induce subcutaneous xenografts from cultured cell lines, 5 × 10^6^ human CRC cells (Colo-205, HCT-116, HT-29, and CACO-2) were injected subcutaneously into a flank of six animals. Tumor diameters were measured weekly with a caliper until reaching a volume of 150 mm^3^, calculated using the following formula: (length × width × width)/2. When all the tumors within a set exceeded the required volume, and the animals were assigned randomly to AD-O51.4-treated and control groups.

#### 4.5.3. AD-O51.4 and TRAIL Administration

AD-O51.4, TRAIL (treated group, 50 mg/kg, based on Rozga et al. [28]) and vehicle (control group) were injected intravenously through the tail vein every other day. Prior to drug administration, the tumor volume and body weight were measured. Mice were carefully observed for the appearance of signs of distress. On day 11, all mice within a set were sacrificed. Blood was collected, and tumors were excised for histopathological and molecular examinations.

### 4.6. Statistical Analysis

Differences between the groups were evaluated using independent-samples t-test, the Mann–Whitney *U*-test and two-way repeated measures ANOVA test. A value of two-sided *p* < 0.05 was considered significant. Analyses were conducted using GraphPad, v8.2 (GraphPad Software, San Diego, CA, USA). Pearson’s correlation coefficients were computed with R package Hmisc and R v3.4.3.

## 5. Conclusions

In our study, we tried to determine the therapeutic effects of AD-O51.4 protein, and proposed a potential mechanism underlying the resistance of CRC cells acquired after prolonged exposure to AD-O51.4.

AD-O51.4 show similar or better cytotoxic activity to human CRC cell lines than the native TRAIL protein in a dose-dependent manner while maintaining its predecessor low toxicity towards normal cells. Further examination using xenograft models revealed that AD-O51.4 inhibits CRC xenograft tumor growth with high efficacy. Immunofluorescence staining showed that as long as AD-O51.4-induced apoptosis is related to the abundance of cell surface receptors, the crucial matter is not the level of individual cell surface receptors but rather their abundance ratios. Experiments on two cell lines with acquired resistance to AD-O51.4 allowed not only to pinpoint HSP60, p53 and Caspase-3 proteins levels alterations as a main source of intracellular resistance, but also to propose the potential mechanism underlying the resistance of CRC cells acquired after prolonged exposure to AD-O51.4.

## Figures and Tables

**Figure 1 ijms-22-03160-f001:**
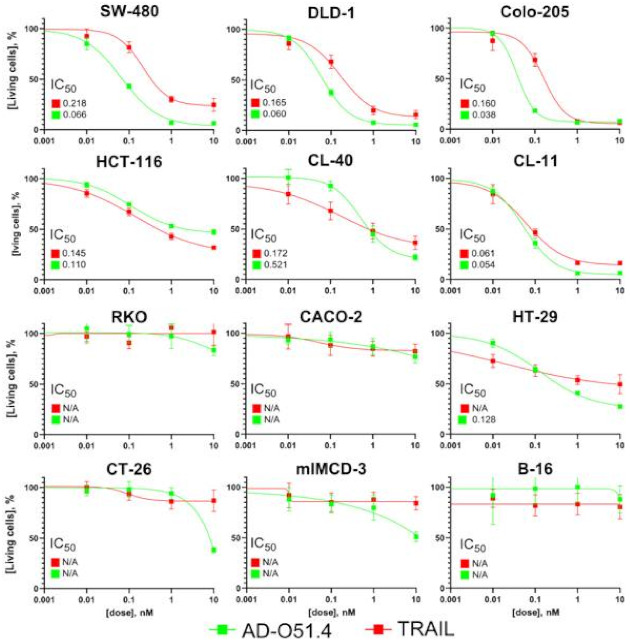
The dose-dependent cytotoxicity of AD-O51.4 and TNF-related apoptosis-inducing ligand (TRAIL) against colorectal cancer (CRC) and mouse (CT-26, mIMCD-3, B-16) cell lines. Results of the MTT assay showing the percentage of living cells together with the IC_50_ values. Cells were seeded at 1.5 × 10^3^ per well and exposed to various concentrations of the indicated agents (0.01, 0.1, 1, and 10 nM) for 48 h. Each bar represents the mean value of eight measurements in two independent experiments. Bars indicate the standard deviation. N/A: not available.

**Figure 2 ijms-22-03160-f002:**
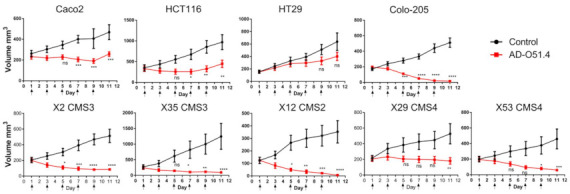
In vivo studies show the decreased growth of the colorectal cancer cell line and patient derived xenografts after AD-O51.4 treatment AD-O51.4 was intravenously administered five times every other day (arrows) at 50 mg/kg. CMSs 1–3 indicate one of the four consensus molecular subtypes (CMSs) of the colorectal cancer [31]. X2, X35, X12, X29 and X53 are internal IDs of patient-derived xenograft (PDX) models. Groups size *n* = 6, each mouse bearing one tumor, ± SEM. Differences between groups were evaluated using two-way repeated measures ANOVA; * *p* < 0.05, ** *p* < 0.01, *** *p* < 0.001, **** *p* < 0.0001; ns—not significant.

**Figure 3 ijms-22-03160-f003:**
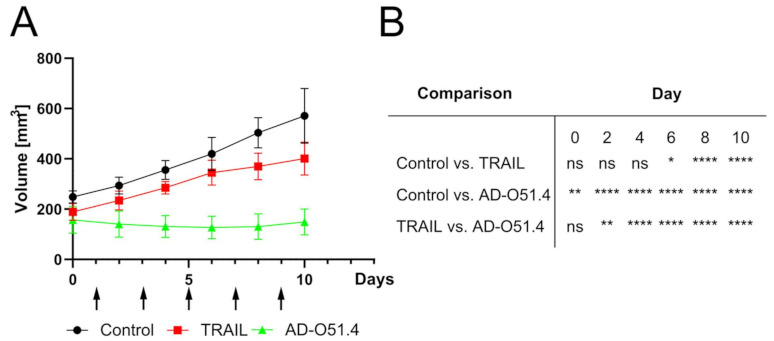
AD-O51.4 inhibits CACO-2 xenograft growth more efficiently than a native TRAIL protein. (**A**) AD-O51.4 and TRAIL were intravenously administered five times every other day (arrows) at 50 mg/kg. Groups size *n* = 6, each mouse bearing one tumor, ± SEM; (**B**) Statistical matrix showing differences between groups were evaluated using two-way repeated measures ANOVA; * *p* < 0.05, ** *p* < 0.01, **** *p* < 0.0001; ns: not significant.

**Figure 4 ijms-22-03160-f004:**
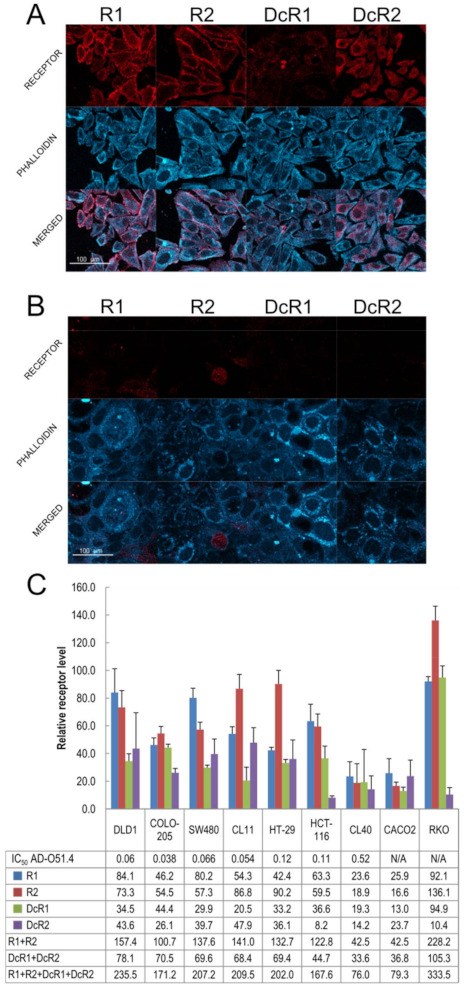
Differences in TNF-related apoptosis-inducing ligand (TRAIL) receptor abundance between nine colorectal cancer (CRC) cell lines. Cell lines were cultured for 48 h in serum-rich media, fixed with paraformaldehyde, and subjected to immunofluorescence staining. Receptors were stained with the indicated antibodies and actin was labeled with phalloidin: (**A**) representative images of receptors in DLD-1, AD-O51.4-sensitive, cell line; (**B**) representative images of receptors in the CACO-2, AD-O51.4-resistant, cell line; (**C**) quantification of receptor density in all examined CRC cell lines. Each bar represents the mean value of three measurements in three independent experiments. Error bars indicate the standard deviation. N/A: not available.

**Figure 5 ijms-22-03160-f005:**
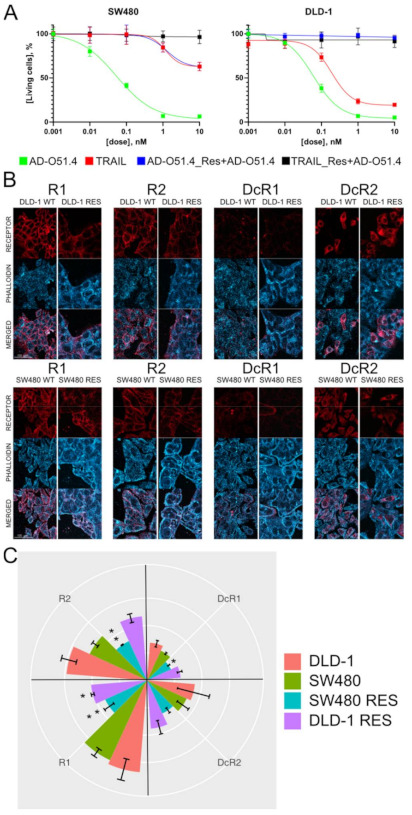
The SW480 and DLD-1 cell lines show the decreased cell surface expression of R1 and R2 TNF-related apoptosis-inducing ligand (TRAIL) receptors after acquiring resistance to AD-O51.4: (**A**) the development of AD-O51.4 resistance in SW480 and DLD-1 AD-O51.4 resistance was generated by culturing SW-480 and DLD-1 cells in the presence of increasing concentrations of AD-O51.4, starting at 0.05 nM. Resistance was examined using the MTT assay cells correlated with TRAIL resistance; (**B**) representative images of receptors in DLD-1 and SW480 cells wild type (WT) and the AD-O51.4-resistant variant (RES); (**C**) quantification of receptor density after the immunofluorescence staining of primary cell lines and cells with acquired resistance. Each bar represents the mean value of three measurements in three independent experiments. Error bars indicate the standard deviation. Differences between groups were evaluated using the Mann–Whitney *U*-test; * *p* < 0.05, ** *p* < 0.01.

**Figure 6 ijms-22-03160-f006:**
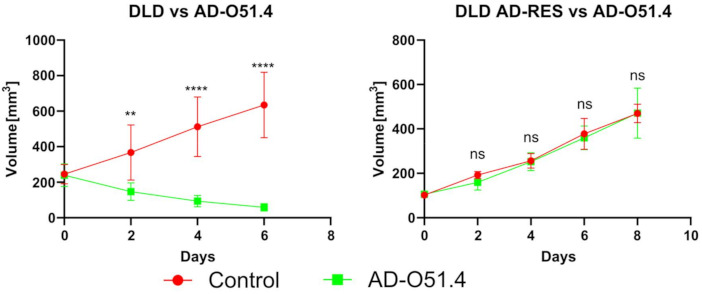
In vivo studies show acquired resistance to AD-O51.4 preservation after xenograft implantation. AD-O51.4 was intravenously administered five times every other day at 50 mg/kg (4 times in the case of the native DLD-1 line). Groups size *n* = 6, each mouse bearing one tumor, ± SEM. ** *p* < 0.01, **** *p* < 0.0001; ns: not significant.

**Figure 7 ijms-22-03160-f007:**
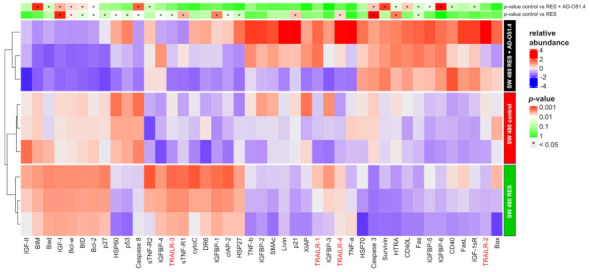
Quantification of apoptosis-related proteins using Human Apoptosis Antibody Array-membrane lysates (500 µg) were processed on the antibody array membrane according to the manufacturer’s recommendations. Blots collected from SW480 native and SW480 cells with acquired resistance to AD-O51.4 cultured with or without AD-O51.4 were subjected to densitometric analysis. The raw blot images files are available at the Figshare repository https://doi.org/10.6084/m9.figshare.12674213.v1.

**Figure 8 ijms-22-03160-f008:**
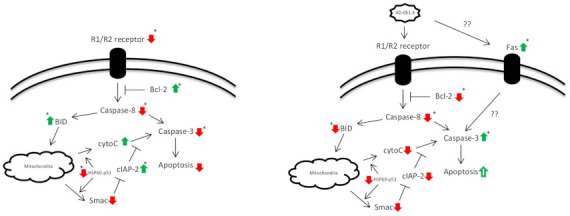
A possible explanation for the extra- and intracellular origins of resistance to AD-O51.4 and TRAIL in SW480 cells. Simplified figure showing the propagation of apoptosis signals within cells. Arrows indicate the increased/decreased expression of specific proteins observed after acquisition of resistance, *—abundance significantly altered (*p* < 0.05) as shown on Figure 5. Left panel: decreased abundance of R1 and R2 surface receptors together with an increased level of Bcl-2 result in a decreased level of Caspase-8 further decreasing Caspase-3 abundance undermining apoptosis. In addition, the decreased abundance of HSP60-p53 complex and increased abundance of cIAP-2 are observed, which suggests inhibition of mitochondrial apoptotic pathway. The same protein alterations were observed by Rozga et al. [28]. Right panel: addition of AD-O54.1 evokes low apoptosis response. Despite an decreased abundance of Caspase-8, increased abundance of Caspase-3 is observed leading to weak apoptosis signal propagation. The Caspase-8-independent Caspase-3 activation could be explained with the observed Fas and FasL increase, furthermore, low BIDlevels preclude mitochondrial pathway involvement in Caspase-3 activation.

## Data Availability

The authors confirm that the data supporting the findings of this study are available within the article. The raw blot images underlying the data presented on Figure 7 are available at the figshare repository under the DOI: 10.6084/m9.figshare.12674213.

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
