# Peer review of "Cytotoxic Efficacy and Resistance Mechanism of a TRAIL and VEGFA-Peptide Fusion Protein in Colorectal Cancer Models"

_ijms, 2021, doi:10.3390/ijms22063160_

Round 1

Reviewer 1 Report

The manuscript is interesting by examining the role of a recently developed chimeric protein (AD-O51.4) based on a TRAIL fused to the VEGFA-derived peptide. The authors have demonstrated that AD-O51.4 showed similar or better in vitro cytotoxic activity than TRAIL in various cancer cell lines from human and murin origin (colorectal, kidney and melanoma cell lines). However the results are not fully analyzed. In addition, the authors have carried out experiments to study the ability of AD-O51.4 to inhibit tumor growth in vivo using xenograft models in NU/J mice and PDX models, but it is not shown whether this effect is similar or stronger than with TRAIL (however one experiment has been carried out in CACO-2 cell line). The authors have also measured the abundance of each cell surface TRAIL receptor to investigate resistance mechanisms, however the various experiments are not well-linked (resistance mechanisms…acquired resistance mechanisms). This point is difficult to read. Overall the data seem not fully exploited and there are major issues.

  • Line 2, title: the title indicates that the manuscript deals with the apoptotic activity of a TRAIL-based protein but no experiments for apoptosis detection have been carried out…the Figure 7 (quantification of apoptosis-related proteins using Antibody Array-membranes does not inform on the proper apoptotic activity of the AD-O51.4…)

  • Introduction: More informations need to be provided to better understand the therapeutic strategy based on the AD-O51.4 chimeric protein (VEGFA peptide…). How the concentration of ADO51.4 has been chosen for the in vivo experiments, there are no explanation to justify this choice.

  • The following sentence is not clear line 74-75: “AD-O51.4 is a first-in-class, …..the first generation of TRAIL agonists” Please explain why.

  • In figure 2, why the antitumor effect of AD-O51.4 has not been compared to the antitumor effect of TRAIL. Fig 1 is a central figure and the TRAIL treated group need to be included.

  • In figure 1, the cytotoxic effect of AD-O51.4 for four cell lines has not been discussed at all in the 2.1 section (HCT116, CL-40, CL-11, HT29)…two of these cell lines are used for the in vivo expreriments (HCT116 and HT29) . IC 50 values have been calculated and

  • Line 146-147: Data are not fully exploited here and the conclusion needs to be revised “..together with the MTT data…. indicate that the extent of apoptosis….” MTT is used as an indicator of cell viability, proliferation and cytotoxicity but it does not inform about apoptosis.

  • Two aspects of resistance is actually studied…”innate resistance” and “acquired resistance” but this is not clearly discuss Figure 4 and Figure 5.

  • Line 167-168 statement need to be revised

  • Figure 4C: results are not fully exploited, ratio DR4/DR5 for example could be exploited. DR5 has a dual role in death and survival signaling…

Minor comments:

Line 15: type II instead of type I

Revised the nomenclature used for the TRAIL receptors…R1, R2, R3, R4…

Line 85: “The effect of AD-O51.4…” à please precise

Line 100: it is not clear whether data from figure 1 are the mean value of eight independent experiments or not (or are octuplicate…)

Author Response

  • Line 2, title: the title indicates that the manuscript deals with the apoptotic activity of a TRAIL-based protein but no experiments for apoptosis detection have been carried out…the Figure 7 (quantification of apoptosis-related proteins using Antibody Array-membranes does not inform on the proper apoptotic activity of the AD-O51.4…)

Response: Based on results from Rozga et al and multiple papers describing that TRAIL and TRAIL-based proteins induce apoptosis we have described our results as apoptosis-based though we agree that these results are indirect with that statement. We replaced referencing to apoptosis with other terms for example cell death / cytotoxic effect.  We also changed the tile to better convey the extent of work presented in the manuscript

  • Introduction: More informations need to be provided to better understand the therapeutic strategy based on the AD-O51.4 chimeric protein (VEGFA peptide…). How the concentration of ADO51.4 has been chosen for the in vivo experiments, there are no explanation to justify this choice.

Response: More accurate data on AD-O51.4 are not available now due to the patent proceeding:  ADAMED SP. Z O.O. (2012). ANTICANCER FUSION PROTEIN. WO/2012/093158. AD-O51.4 used concentration are based on Rozga et al work ( Int J Cancer. 2020 Aug 15;147(4):1117-1130. doi: 10.1002/ijc.32845). This information is now included in the methods section.

  • The following sentence is not clear line 74-75: “AD-O51.4 is a first-in-class, …..the first generation of TRAIL agonists” Please explain why.

Response: the sentence was deleted.

  • In figure 2, why the antitumor effect of AD-O51.4 has not been compared to the antitumor effect of TRAIL. Fig 1 is a central figure and the TRAIL treated group need to be included.

Response: As mentioned in paper, this work is more in depth evaluation of AD-O51.4 treatment of CRC. First paper by Rozga et al. describing this protein has a screening character, where authors evaluated AD-O51.4 toxicity towards various cancer types, including colorectal cancer cell lines LS1034 and Colo205 grown as xenografts as well as one CRC PDX. They included native TRAIL control in in vivo experiments showing AD-O51.4 superiority. Because of that, we have resigned from using TRAIL control, only incorporating it when results were unclear or unexpected like with Caco-2 xenograft.

  • In figure 1, the cytotoxic effect of AD-O51.4 for four cell lines has not been discussed at all in the 2.1 section (HCT116, CL-40, CL-11, HT29)…two of these cell lines are used for the in vivo expreriments (HCT116 and HT29) . IC 50 values have been calculated and

Response: We incorporated the sensitivity grading, highly, moderate and no sensitive, based on the IC50 values and assigned tested cell lines accordingly.

  • Line 146-147: Data are not fully exploited here and the conclusion needs to be revised “..together with the MTT data…. indicate that the extent of apoptosis….” MTT is used as an indicator of cell viability, proliferation and cytotoxicity but it does not inform about apoptosis.

Response: We now do not refer to the apoptosis since it was not evaluated directly in our work

  • Two aspects of resistance is actually studied…”innate resistance” and “acquired resistance” but this is not clearly discuss Figure 4 and Figure 5.

Response: We changed figure 5 description so now it is more distinguishable. Description in section 2.3 clearly states that at this point we are evaluating acquired resistance.

 Line 167-168 statement need to be revised

Response: Statement revised. 

  • Figure 4C: results are not fully exploited, ratio DR4/DR5 for example could be exploited. DR5 has a dual role in death and survival signaling…

Response: As could be read in PMID:29048428, it is true that DR5/TRAIL-R2/R2 receptor has an dual role, but even there its pro-survival part is only fractional against pro-apoptotic. We admit, that there could be a possibility where pro-survival role of R2 receptor play significant role in AD-O51.4 treatment response, but our results for 8 out of nine CRC cell lines tested (besides RKO) underline that high levels of R2 receptor correlate with cell death extent.

Minor comments:

Line 15: type II instead of type I

Corrected

Revised the nomenclature used for the TRAIL receptors…R1, R2, R3, R4…

In introduction section we have stated that there are few naming systems for TRAIL receptors, we have used shortened from of TRAIL-R1, TRAIL-R2….. and used it through hole paper. This information is included in the introduction.

Line 85: “The effect of AD-O51.4…” à please precise

Response: the “cytotoxic effects” is included  

Line 100: it is not clear whether data from figure 1 are the mean value of eight independent experiments or not (or are octuplicate…)

Response: The description was clarified: “Each bar represents the mean value of eight technical measurements in one experiment”

Reviewer 2 Report

Kopczynski et al. showed that AD-O51.4 protein, a TRAIL fused to the VEGF-originating peptide, inhibited cell proliferation and xenograft tumor growth in human colorectal cancer (CRC) cell lines and investigated the acquired resistance mechanism(s) using CRC cells, SW480, and DLD-1. They revealed that the attenuation of AD-O51.4’s target, R1 and R2 TRAIL receptors, occurred in resistance clones of SW480 and DLD-1, and concluded that the decrease in cell surface protein of R1 and R2 TRAIL receptors conferred resistance to AD-O51.4.

Although the manuscript included interesting results, there are a few points that require further clarification.

The manuscript should be accepted for publication.

Comments:

  1. The authors did not detect the induction of apoptosis, although inhibition of cell proliferation was determined by AD-O51.4 treatment. Apoptosis induction should be determined in vitro and in vivo.
  2. (Line 90 - 91) The authors should describe the reason why AD-O51.4 did not inhibit the three mouse cell lines.
  3. (Figure 1) Which are the mouse cell lines? Please add the information about the cell lines in the figure legend and materials and method.
  4. (Line 137 - 138) The authors should add the reason why immunofluorescence staining is superior to western-blotting in the detection of TRAIL receptors. Please add a reference(s) to support this evidence.
  5. (Figure 1 and 4c) The expression levels of the TRAIL receptors in RKO cells are relatively higher than those in the other cell lines, although AD-O51.4 did not inhibit cell proliferation in Figure 1. Please add a description in the Results and Discussion section.
  6. (Line 257) There is no Figure 3C.

Author Response

  1. The authors did not detect the induction of apoptosis, although inhibition of cell proliferation was determined by AD-O51.4 treatment. Apoptosis induction should be determined in vitro and in vivo.

As already pointed in the response to the Referee#1 we agree that our data do not show the induction of apoptosis directly. However, the apoptotic cell death of CRC cell lines also grown as xenografts induced by AD-O51.4 has already been shown by Rozga et al. We To better convey the extent of work presented in the manuscript we removed the apoptosis from the title.

  1. (Line 90 - 91) The authors should describe the reason why AD-O51.4 did not inhibit the three mouse cell lines.

Response: More information on mouse TRAIL receptors and speculations on why AD-O51.4 in not cytotoxic in murine systems are included.

  1. (Figure 1) Which are the mouse cell lines? Please add the information about the cell lines in the figure legend and materials and method.

Response: Mouse cell lines are now listed in the legend and information on their culturing condition has been included in the methods section.

 (Line 137 - 138) The authors should add the reason why immunofluorescence staining is superior to western-blotting in the detection of TRAIL receptors. Please add a reference(s) to support this evidence.

Response: We have changed “superior” to “more accurate” and rewritten this sentence to support this statement.

  1. (Figure 1 and 4c) The expression levels of the TRAIL receptors in RKO cells are relatively higher than those in the other cell lines, although AD-O51.4 did not inhibit cell proliferation in Figure 1. Please add a description in the Results and Discussion section.

Response: that finding is noticed and now described in results section (line 127-128). In the discussion we have proposed, that RKO resistance could be of intracellular origin (Line 226-228).

(Line 257) There is no Figure 3C.

Response: Changed to 4C

Round 2

Reviewer 1 Report

Major concerns :

Figure 1 : Authors have now indicated that the data from Figure 1 are from one experiment only even though it is the mean value of eight technical measurement…it is difficult to build conclusions on one experiment.

Figure 4 :The results from Figure 4 are not fully exploited. As the authors mentioned in the discussion « the levels of DcR1 and DcR2 are crucial ». However DcR1 and DcR2 levels from Figure 4C are not interpreted at all. Figure 4C mentionned two cell lines DLD1 and Caco2 cell lines but more cell lines were used in Figure 4C : 7 other cell lines were studied for TRAIL receptor abundance and no results are exploited

Discussion : line 212 the conclusion « AD-051.4 delayed growth of tumors derived from all cell lines tested », according to the figure 2 it seems not to be the case for HT29 cells…

Minor concerns :

  • Line 14 is still incorrect : « type II » instead of « type I »
  • Line 38 replace « digested » by « cleaved »

Author Response

Figure 1 : Authors have now indicated that the data from Figure 1 are from one experiment only even though it is the mean value of eight technical measurement…it is difficult to build conclusions on one experiment.

Response: We have incorporated data from additional independent experiments (8 technical measurements each) and updated graphs and IC50 values accordingly.

Figure 4 :The results from Figure 4 are not fully exploited. As the authors mentioned in the discussion « the levels of DcR1 and DcR2 are crucial ». However DcR1 and DcR2 levels from Figure 4C are not interpreted at all. Figure 4C mentionned two cell lines DLD1 and Caco2 cell lines but more cell lines were used in Figure 4C : 7 other cell lines were studied for TRAIL receptor abundance and no results are exploited

Response: Results are now better exploited, we calculated Perason’s correlation coefficient showing that the sensitivity to AD-051.4 correlates with combined TRAIL receptor abundances.

Discussion : line 212 the conclusion « AD-051.4 delayed growth of tumors derived from all cell lines tested », according to the figure 2 it seems not to be the case for HT29 cells…

Response: Changed to" derived from 4 out of 5 cell lines tested"

Minor concerns :

  • Line 14 is still incorrect : « type II » instead of « type I »

changed

  • Line 38 replace « digested » by « cleaved »

changed

.